# Are Aminoglycoside Antibiotics TRPing Your Metabolic Switches?

**DOI:** 10.3390/cells13151273

**Published:** 2024-07-29

**Authors:** Alfredo Franco-Obregón, Yee Kit Tai

**Affiliations:** 1Department of Surgery, Yong Loo Lin School of Medicine, National University of Singapore, Singapore 119228, Singapore; 2Institute of Health Technology and Innovation (iHealthtech), National University of Singapore, Singapore 117599, Singapore; 3BICEPS Lab (Biolonic Currents Electromagnetic Pulsing Systems), National University of Singapore, Singapore 117599, Singapore; 4NUS Centre for Cancer Research, Yong Loo Lin School of Medicine, National University of Singapore, Singapore 117599, Singapore; 5Competence Center for Applied Biotechnology and Molecular Medicine, University of Zürich, 8057 Zürich, Switzerland; 6Department of Physiology, Yong Loo Lin School of Medicine, National University of Singapore, Singapore 117593, Singapore

**Keywords:** antimicrobial resistance, mitohormesis, magnetic fields, adipogenesis, regenerative medicine, muscle–adipose crosstalk, myokines, insulin resistance, diabetes, inflammation

## Abstract

Transient receptor potential (TRP) channels are broadly implicated in the developmental programs of most tissues. Amongst these tissues, skeletal muscle and adipose are noteworthy for being essential in establishing systemic metabolic balance. TRP channels respond to environmental stimuli by supplying intracellular calcium that instigates enzymatic cascades of developmental consequence and often impinge on mitochondrial function and biogenesis. Critically, aminoglycoside antibiotics (AGAs) have been shown to block the capacity of TRP channels to conduct calcium entry into the cell in response to a wide range of developmental stimuli of a biophysical nature, including mechanical, electromagnetic, thermal, and chemical. Paradoxically, in vitro paradigms commonly used to understand organismal muscle and adipose development may have been led astray by the conventional use of streptomycin, an AGA, to help prevent bacterial contamination. Accordingly, streptomycin has been shown to disrupt both in vitro and in vivo myogenesis, as well as the phenotypic switch of white adipose into beige thermogenic status. In vivo, streptomycin has been shown to disrupt TRP-mediated calcium-dependent exercise adaptations of importance to systemic metabolism. Alternatively, streptomycin has also been used to curb detrimental levels of calcium leakage into dystrophic skeletal muscle through aberrantly gated TRPC1 channels that have been shown to be involved in the etiology of X-linked muscular dystrophies. TRP channels susceptible to AGA antagonism are critically involved in modulating the development of muscle and adipose tissues that, if administered to behaving animals, may translate to systemwide metabolic disruption. Regenerative medicine and clinical communities need to be made aware of this caveat of AGA usage and seek viable alternatives, to prevent contamination or infection in in vitro and in vivo paradigms, respectively.

## 1. Background

Muscle and adipose are the body’s two most critical tissues for establishing systemic metabolic balance. These two tissues synchronize each other’s metabolic status via their mutual exchange of paracrine factors—myokines and adipokines, respectively—that have both trophic and metabolism-modulating activities [1,2,3]. Ultimately, this system of paracrine crosstalk expands beyond the realm of muscle and fat to encompass the other tissues of the body [4], including the liver [5,6], heart [6,7], bone [8,9], and various inflammation-regulating systems [10,11]. 

Skeletal muscle is the largest tissue mass in healthy individuals and, by design as a weight-supporting tissue, is constantly mechanically loaded by the unyielding force of gravity. Weight-bearing provides a basal level of muscle mitochondrial activation (to sustain posture) that sets the enzymatic backdrop for the muscle secretome response [12]. Consequently, the experiences of prolonged periods of exposure to microgravity and bed rest share many of the same physiological disruptions [13]. Unlike skeletal muscle, adipose tissue is not organized within the body to resist the force of gravity, whereas skeletal muscle is organized for the maintenance of a functional posture for the execution of daily activities. Skeletal muscle is hence the initial and principal point for modulation by gravitational/mechanical loading. Consequently, in non-obese individuals, the influence of skeletal muscle presides in establishing the balance between myokine–adipokine exchange.

If muscular interaction with gravity is further augmented by the addition of movement, otherwise known as exercise [4,14], the muscle secretome response may become sufficiently augmented to attenuate the inflammatory status of white adipose. A reduction in white adipose inflammation, in turn, would then be reflected as a reduction in the inflammatory status of the adipose secretome [15]. Specifically, white adipose phenotypically “beiges” when subjected to anti-inflammatory myokines (exerkines), characterized by adipose mitochondrial uncoupled respiration and non-shivering thermogenesis [15,16]. Adequately beiged adipose tissue, in turn, reciprocates with the release of brown adipose-associated adipokines, or batokines, that reinforce the anti-inflammatory status of muscle [12,17,18,19]. Activating muscle via “exercise” hence improves the systemic metabolism and inflammatory status by first targeting adipose tissue in a way that later transcends to the entire system [10,11].

On the other hand, excessive caloric intake in combination with a sedentary lifestyle creates a physiological scenario via which the inflammatory status of adipose tissue is left relatively unchecked under the reduced influence of the muscle secretome. This results in an accumulation of inflamed white adipose tissue in the thoracic subcutaneous and visceral deposits, as well as ectopic adipose deposition in the liver, heart, and skeletal muscle, that in turn, assumes detrimental metabolic dominance by augmenting the release of pro-inflammatory adipokines into the systemic circulation [15], instilling a state of systemic inflammation [20]. Obesity-associated systemic inflammation, in turn, compromises muscle development [21] as well as attenuating muscle’s ability to metabolically adapt to exercise [22] that, in turn, undermines muscle’s secretome modulation of adipose [2,3,4,10,15]. Key in tipping this delicate tissue balance in favor of a healthful metabolic status is the attenuation of white adipose inflammation via the activation of muscle with exercise.

Fortunately, key aspects of the exercise muscular metabolic cascade remain intact during systemic inflammation. Despite the existence of systemic insulin resistance, acute exercise must still stimulate mitochondrial respiration, increasing the demand for oxygen and metabolic substrates for energy production [23]. The exhaustion of molecular oxygen by respiring mitochondria during exercise is a key factor in this innate muscular tolerance to systemic inflammation that has been shown to promote cell survival [24]. If the execution of exercise results in an insufficiency of oxygen to respiring mitochondria [25], then mitochondriogenesis and mitohormetic adaptations may be stimulated [26,27]. Inherent to this mitohormetic response is an oxidative shift in skeletal muscle mitochondrial substrate utilization towards more energy-enriched fatty acids [28,29] that ultimately assists in reducing adipose reserves. Hence, the vicious cycle set in motion by white adipose inflammation can be interrupted, or potentially reversed, by exercise. Exercise re-establishes an anti-inflammatory muscle secretome response that reinstates systemic metabolic balance [4,14].

Critically, aminoglycoside antibiotics (AGAs) have been shown to block the ability of muscles to adapt to exercise [30,31,32], particularly towards an oxidative phenotype [33,34] (also see Table 1). This detrimental aspect of AGAs can have critical implications for human health status. Oxidative muscle is associated with the greatest metabolic benefits [32,33,35,36], as well as possessing stronger metabolism-stabilizing secretomes [37,38]. Evidence also exists that the AGAs interfere with the transition of white adipose tissue into the beiged state [39] or preferentially stabilize the white adipose phenotype [40]. These combined antagonisms of the AGAs over both muscle and adipose determination could severely disrupt human metabolic adaptations to exercise, as depicted in Figure 1. 

Streptomycin (100 μg/mL), but not penicillin (150 U/mL) or amphotericin B (5 μg/mL), was shown to preferentially block protein accretion into primary myogenic cultures under identical conditions [41], indicating that the aminoglycosides exert a distinct form of developmental disruption. Indeed, subsequent studies revealed that the AGAs exhibit a form of developmental antagonism that is unique amongst the other antibiotic classes, since it entails the immediate interference of ion channel-mediated signal transduction arising from diverse biophysical stimuli, such as mechanical force, magnetism, light, or temperature, into developmental responses (see Section 3: TRP Channels Transduce Biophysical Stimuli into Mitohormetic Responses), rather than the more protracted inhibition of protein translation commonly attributed to a wide scope of antibiotics [42]. As a point of distinction, AGAs need only to be present during the presentation of the biophysical stimuli to effectively inhibit developmental signal transduction, and therefore, in this respect, their requisite time course of action can be shown to be remarkably brief [34]. Speed of action is particularly evident when the experiment is specifically designed with this intent in mind. The duration of exposure to the AGAs has varied at the discretion of the researcher and ranged from days [30], to 10 s of minutes [43], to instantaneous, as when examined cation channel activity is measured via the patch electrode [44]. For instance, Iversen et al. recently demonstrated that streptomycin applied during a brief exposure to light, magnetic fields, or combined light and magnetic fields precluded the induction of cell proliferation normally induced by analogous exposure, whereas streptomycin applied immediately after exposure did not [43] (Figure 2). As the duration of the electromagnetic exposure was only 15 min and cell analyses were conducted 24 h following exposure, the difference in time spent in streptomycin (30 min) was relatively negligible and unable to account for the antagonism observed when applied during exposure. Although commonly overlooked in clinical literature, this aspect of AGAs could have profound developmental consequences.

**Table 1 cells-13-01273-t001:** AGA antagonism of exercise effects.

AGA Antagonism of Exercise Effects						
Year	Tissue/Cell Type	Animal	Context	AGA (dose)	Relevant Findings	Ref.
2003	Muscle (Tibialis Anterior)	Rat (female Sprague Dawley rats, 3 months of age)	Electrical Eccentric Contractions (Stretch-induced depolarizations in Tibialis Anterior muscles)	Streptomycin (4 g/L: drink water) Streptomycin given in drinking water 6 days prior to Electrical Eccentric Exercise and terminated immediately before analysis. Other SAC Blockers: GdCl_3_ (10 μM)	Streptomycin blocked accommodation of eccentric contraction-induced depolarization and c-*fos* increase. Streptomycin blocked adaptive muscle hypertrophy after eccentric contraction muscle training. In vitro GdCl_3_ augmented streptomycin effects. Implicated SACs in response.	[30]
2006	Muscle (Tibialis Anterior)	Rat (female Sprague Dawley rats, 3 months of age)	Electric Eccentric Contractions (stretch-induced activation of Akt-mammalian target of rapamycin-p70 S6 kinase (p70^S6K^) signaling pathway)	Streptomycin (4 g/L; drinking water) Streptomycin given in drinking water 6 days prior to Electrical Eccentric Exercise and terminated immediately before analysis. Other SAC Blockers: GdCl_3_ (80 μmol/kg IV).	Streptomycin blocked rise in Akt and phosphorylation of p70^S6K^ associated with eccentric contraction. Streptomycin also blocked the phosphorylation of GSK-3b and S6 in response to muscle contraction. Both streptomycin and GdCl_3_ blocked contraction-induced hyperpolarization of the muscle resting membrane potential without altering force production. Implicated SACs in response.	[35]
2009	Muscle (Dorsiflexors)	Rabbit (New Zealand White)	12 bouts Electric Eccentric Exercise over four weeks	Streptomycin (800 mg/mL injected daily for 40 days intramuscularly at a dose of 300 mg/kg body weight).	Streptomycin eliminates functional adaptations of muscle after eccentric exercise. Implicated SACs in response.	[32]
2015	Muscle (cell line)	Mouse muscle cell line (C2C12)	Electrical chronic low-frequency stimulation of engineered muscle	Streptomycin (added to culture media at a concentration of 100 μg/mL).	Streptomycin inhibited fast-to-slow phenotype shift of C2C12-engineered muscle as indicated by depressions of contraction-induced fatigue resistance, GLUT4 levels, and the mitochondrial proteins succinate dehydrogenase and ATP synthase. Implicated SACs and voltage-gated Ca^2+^ and K^+^ channels in response.	[33]
2015	Bone (Tibialis Anterior)	Rat (F344; seven-week-old males)	Electrical muscle stimulation for 30 min/day for 6 days/week for one week.	Streptomycin (4 g/L in drinking water administered for 6 days prior to surgery).	Streptomycin did not induce bone loss, but attenuated protection offered by electrically stimulated muscle contraction against disuse-induced trabecular bone loss in muscle-denervated rats.	[36]
2018	Muscle (Tibialis Anterior)	Rat (Fischer 344; seven-week-old males)	Electric Eccentric Contractions	Streptomycin (4 g/L in drinking water administered for 6 days prior to Electrical Eccentric Exercise).	Streptomycin reduced eccentric contraction-induced Evans blue dye uptake by muscle fibers, but did not increase in cross-sectional area or decrease in roundness.	[45]

Abbreviations: SAC: stretch-activated channel; aminoglycoside antibiotic (AGA).

## 2. Exercise Mitohormetic Adaptations Are Antagonized by AGAs

The increase in mitochondrial respiration that, by necessity, accompanies exercise serves as the basis for the metabolic adaptations that are initially instilled in skeletal muscle and later spread to the rest of the organism. The ensuing energy deficit and oxidative stress instigates a compensatory transcriptional response within muscle that is designed to improve metabolic efficiency, stimulates mitochondriogenesis, enhances the antioxidant defenses of mitochondria, and promotes cell survival in the face of oxidative stressors [27,38,46]. This adaptive process is known as mitohormesis [47,48]. Via mitohormetic principles, low levels of oxidative stress promote cell survival, whereas excessive oxidative stress results in cell demise. Secretome activation is one of the ensuing response limbs of the mitohormetic cascades invoked by exercise [47,49], which is then translated to the rest of the organism [14,50,51]. Because of the mitohormetic adaptations of muscle, exercised muscles increase their oxidative capacities and become better “secreters” in response to exercise as well as during moments of rest [52]. Muscle mitohormetic adaptations hence translate into improved basal systemic metabolism and overall health. Nonetheless, they are susceptible to environmental confounders.

AGAs may be one such class of mitohormetic disruptor. AGAs have been shown to block myogenic secretome release [53] as well as secretome action [52]. Importantly, any circumstance, or agent, that would interfere with muscle’s ability to respond to exercise would leave adipose tissue unchecked and facilitate systemic inflammation to preside. AGAs are hence valid candidates for culpable agents that may lead to metabolic disruption by interfering with the mitohormetic effects of exercise. Evidence exists that AGAs exert their antagonism of mitohormetic adaptations by interfering with biophysical signal transduction by transient receptor potential (TRP) channels [34]. 

## 3. TRP Channels Transduce Biophysical Stimuli into Mitohormetic Responses

Underappreciated is the fact that biophysical stimuli are fundamental developmental determinants [54,55]. At the cellular level, TRP channels are responsible for transducing biophysical stimuli into developmental responses [56]. Additionally, for a biophysical stimulus to be transduced into an enzymatic cascade by a TRPC channel complex (composed of tetramers of TRPC channel subunits), it need not occur in isolation but may be combined with other biophysical stimuli of distinct modalities simultaneously impinging upon the cell and transduced by other TRPC channel complexes or subunits. The manner in which TRPC channel complexes are capacitated for transducing diverse biophysical stimuli is via a process of heterologous multimerization (heterotetramers) of different “C” subunits possessing sensitivities to distinct forms of biophysical stimuli [57,58]. TRPC subunit heteromultimerization effectively unifies diverse gating sensitivities into a single-channel complex [59] to produce a unified cell response. In this manner, TRPC channel complexes subserve the role of integrators of diverse biophysical stimuli of developmental consequence. In summary, heteromultimerization is the modus operandi of TRP subunits, and cellular signal integration is the benefit it confers.

The TRPC1 subunit appears to be pivotal for the integration capacity of a TRP channel complex to be realized. The TRPC1 subunit regulates the activity of the other TRPC family members by forming heteromultimers with them [57,58]. The TRPC1 subunit also heteromultimerizes with the subunits of other TRP families to further extend the range of sensitivities of the channel complex to biophysical stimuli [60,61]. Accordingly, activation modes associated with the TRPC1 subunit include mechanotransduction [62,63,64,65], magnetotransduction [34,66,67,68], phototransduction [69,70,71], thermosensation [72], mitogen activation [57,58], phospholipid activation [62,73], calcium-sensing/SOCE (store-operated calcium entry) [74], redox sensing [75,76,77,78], and mitogen sensing [57,58]. TRP channel-mediated stimuli signal transduction must ultimately be transformed into mitohormetic adaptations.

Mitochondria provide the molecular energy (ATP) that underlies any cellular response to an impinging stimulus, noxious or beneficial. Mitochondria are thus fundamental to the adaptive mitohormetic adaptations that govern our health and lifespan [48,79]. Capacitating mitohormetic adaptations, and in accordance with their role as biophysical stimuli receptors, the activation of TRPC channels is relayed to the mitochondria. Mitochondrial responses are hence a common response limb of TRPC channel activation [34,80]. Mechanistically, the TRPC family members are regulated by, and responsible for, sustained calcium entry [81,82,83]. TRPC channel-mediated calcium entry also modulates mitochondrial function [84,85,86]. Conversely, mitochondrial-derived reactive oxygen species (ROS) regulate TRPC channel activity [75,76,77]. The communication between TRPC channels and the mitochondria is thus two-way [87] and serves to mutually consolidate cellular responses to developmental stimuli.

## 4. TRP-Mediated Mitohormetic Responses Are Antagonized by AGAs

TRP channels are widely distributed amongst the different tissues of the body [57,88] and have been implicated in diverse developmental programs [89,90]. Noteworthily, certain TRP channel families have been implicated in the mitohormetic adaptations of muscle and adipose tissues. Critically, AGAs possess the ability to block calcium entry via these TRP channels (Table 2) to preclude downstream mitochondrial response cascades in these tissues [34,39]. The antagonism of these TRP channels by the AGAs will thus limit the capacity of muscle and adipose tissues to adapt to exercise and thus positively modulate systemic metabolism. The mechanism of acute antagonism of these TRP channels by the AGAs goes beyond the more commonly ascribed inhibition of the protein synthetic cellular machinery shared by most antibiotics [42]. Specifically, the AGAs disrupt cellular signal transduction by physically impeding Ca^2+^ permeation through TRP channels. This is achieved by virtue of their high affinity to bind within, and thereby sterically obstruct, the pore region of TRP channels [44,61,91,92,93]. Given the structural similarity of the pore regions between certain TRP channel classes [60,94], AGA antagonism may represent a general feature of multiple TRP channel classes (also see Section 10: AGA antagonism of other TRP channels). To be capable of responding to biophysical stimuli, TRPC channels are likely required to interact with other subcellular components, rather than to directly perceive and respond to biophysical stimuli per se [95]. The integrative nature of TRPC1 hence holds broad developmental importance that can be undermined by AGAs. The AGAs would hence interfere with the establishment of muscle and fat TRP channel-mediated metabolic adaptations arising from exercise or other forms of biophysical stimulation.

## 5. Gravity and the Geomagnetic Field Are Unyielding Biophysical Developmental Forces 

The Earth’s gravitational and geomagnetic fields are the two biophysical forces to which we are always exposed on Earth. Ample evidence exists demonstrating that both gravity and the geomagnetic field are authentic and fundamental developmental forces [96]. Most importantly, gravitational and geomagnetic fields modulate oxidative muscle development, also known as “antigravity” muscle [38,96,97,98]. By contrast, other forms of biophysical stimuli such as temperature and light change diurnally in the natural environment, as well as being intervened with at the hands of man. Notably, gravitational force, manifested as weight bearing [60,62,65], and magnetic fields [38,96,99] are amongst the biophysical forces transduced by TRPC1-containing channels. TRPC1 is ubiquitously expressed [39,57,59,88] and, alone or in combination with other TRP channel subunits, is involved in the transduction of both mechanical forces [63,64,92,100,101] and magnetic fields [34,53,66,67,68,102]. The mechanistic interplay between the mechano- and magneto-sensitive modes of TRPC1 has been previously postulated and the implications discussed [96,103]. Most critically for the context of the current discussion, TRPC1 function is essential for oxidative/antigravity muscle development (see Section 6: Myogenesis), and since oxidative muscle is constantly activated by the force of Earth’s gravity [97,98], their blockage by AGAs would have profound physiological ramifications.

The Earth’s standing gravitational field is a constant developmental imperative. Yet, due to its ever-present nature, its relevance to human health is commonly misunderstood by the layperson. In practice, however, interaction with the Earth’s gravitational field is under our own control, predominantly implicates muscle, and is modulated by physical movement that is otherwise referred to as exercise. Mechanical force is transmitted at the organismal level, from cells to tissues, by gravitational mechanical loading of the musculoskeletal frame. Exercise can hence be classified as movement under the force of gravity and/or muscle contraction against the inertia of objects weighted by gravity. In hominoids, the assumption of an upright posture fostered the development of the antigravity, or oxidative, musculature [97]. Conversely, oxidative muscle is lost under conditions of reduced gravitational force, such as during manned space flight [104,105]. Oxidative muscle exhibits the greatest resistance to fatiguing and is associated with systemic metabolic balance and longevity [97,106]. Oxidative muscle is exemplified by the soleus muscle that runs underneath and along either side of the calve musculature of the lower leg. The soleus allows us to maintain a standing posture under the constant force of gravity and therefore, must be capable of sustaining contractions for long periods of time without fatiguing. In particular, the activation of the soleus muscle has been shown to be intimately associated with systemwide metabolic health [107]. Therefore, the basic act of maintaining a standing posture, or walking, under the constant force of gravity is important in establishing basal metabolism. Therefore, exercise, in its most fundamental form, is a state of gravitational loading of metabolic consequence.

On the other hand, our physiological interaction with magnetism is a bit more enigmatic. Outside of the Earth’s standing geomagnetic field, which varies in inclination and amplitude at different regions of the planet, and intentional magnetic therapies, exposure to exogenous magnetism, both man-made (power lines, etc.) and natural (lodestones made of magnetite and geomagnetic storms, etc.) forms, is largely haphazard [108,109]. Nonetheless, evidence is accumulating indicating that appropriately conceived magnetic therapies can have beneficial consequences with regards to human health [38,110,111]. Indeed, unequivocal evidence exists that electromagnetic fields are an authentic developmental force [96,112], as shielding cells from all magnetic fields halts their development [34]. Notably, the development of the soleus muscle is preferentially promoted by sustained mechanical loading [113] as well as certain forms of magnetic field exposure [114]. In terms of stimulus transduction, TRPC1-mediated calcium entry is stimulated by mechanical loading [115,116] as well as magnetic field exposure [34,114] as a prerequisite for soleus development. Yet, neither magnetism nor mechanical forces vicariously act in the form of the other. That is, the magnetic exposure paradigms that have been implicated in TRPC1 activation do not inadvertently produce mechanical stress [34], and likewise, the mechanical stimuli commonly used in in vitro studies should not induce ionic fluxes of a sufficient magnitude to produce appreciable magnetic fields in the micro-milliTesla range [109,117,118,119]. Notably, AGAs have been shown to antagonize both the mechanical and electromagnetic cellular responses commonly attributed to TRPC1 and, when explicitly tested, can be emulated by established blockers of mechanosensitive or TRPC1 channels (Table 2). These responses can be further extrapolated to the AGA-mediated inhibition of exercise-induced adaptations (Table 1). For historical reasons, streptomycin is the most thoroughly tested of the AGAs, although neomycin may demonstrate a higher efficacy of blocking for both the voltage-gated [120] and stretch-activated/TRP (Table 2) Ca^2+^-permeable channels. The inhibition of TRPC1-mediated Ca^2+^ entry by AGAs will hence simultaneously counteract the actions of our two most fundamental developmental forces, gravitational mechanical loading and magnetism [54,121,122,123], for oxidative skeletal muscle that is intimately linked to systemic metabolic balance.

**Table 2 cells-13-01273-t002:** AGA antagonism of stretch-activated/mechanosensitive channel.

AGA Antagonism of Stretch-Activated/Mechanosensitive Cation Channels							
Year	Tissue	Animal	Context	Modulator	Channel	Findings	Ref.
1982	Sacculus Hair Cells	Bullfrog (Rana catesbeiana)	Hair Bundle Deflection	Dihydrostreptomycin (~10 μM near bundle calculated from 400 mM within iontophoretic electrode)	SACs	Dihydrostreptomycin blocked sinusoidal deflection-activated currents within the hair bundle/stereocilia.	[124]
1985	Vestibular Hair Cells	Chicken	Hair Bundle Deflection	Streptomycin (500 μM) Neomycin (750 μM)	SACs	Streptomycin and meomycin blocked mechanosensitive/deflection-activated Ca^2+^ currents. Streptomycin was a more potent blocker than neomycin.	[125]
1989	Sacculus Hair Cells	Bullfrog (Rana catesbeiana)	Otolithic Membrane Displacement	Dihydrostreptomycin (500 μM) Gentamicin (500 μM)	SACs	AGAs reversibly blocked displacement-activated transduction channels in the otolithic membrane.	[126]
1993	Muscle	Chicken	General Review of SAC Modulators (Patch Clamp Recordings)	0 V_m_ KDs: Neomycin (2 pM) Streptomycin (21 pM) Ribostamycin (32 pM) Dibekacin (48 pM) Kanamycin (55 pM)	SACs	Aminoglycoside antibiotics block SAC single-channel currents in a dose- and voltage-dependent manner in the sequence neomycin > streptomycin > ribostamycin > dibekacin > kanamycin. GdCl_3_ is also identified as an SAC blocker. The dissociation constants of the aminoglycoside antibiotics for SACs are lower than for other Ca^2+^ or K^+^ channel classes.	[44]
1994	Heart Ventricular Myocytes	Guinea pig (Cavia porcellus)	Stretch Ca^2+^ Transients	Streptomycin (40 μM) Penicillin (40 μM)	SACs	Streptomycin blocked stretch-induced Ca^2+^ transients but had no effect on L-type voltage-gated Ca^2+^ currents. Streptomycin exhibited faster blocking action than penicillin and was inferred from published literature to be more specific for SACs than GdCl_3_.	[127]
1996	General Review	Vertebrates and C. elegans	General Review of SAC Pharmacology	AGAs (2–60 μM) Table 2: IC_50_ values of different AGAs on different channel classes	SACs	General review of SAC modulators including AGAs and GdCl_3_. IC_50_ values are generally lowest for SACs compared to other cation channel classes.	[128]
1996	Muscle (Flexor Digitorum Brevis)	Mouse (C57BL/6)	Suction via Patch Electrode During Single-Channel Recordings	Dihydrostreptomycin (50–100 μM) Neomycin (50–100 μM) Gentamicin (1000–2000 μM) Amikacin (100–400 μM) Streptomycin (500–800 μM) Kanamycin (800–100 μM)	SACs	AGAs reversibly blocked stretch-activated channel currents at discrete subconductance levels. Aminoglycoside antibiotic blocking potency: dihydrostreptomycin, neomycin > gentamicin > amikacin, streptomycin > kanamycin. Table 1: KDs of AGAs.	[91]
2000	Muscle (Tibialis Anterior)	Rat (Sprague Dawley)	Electrical Eccentric Contraction	Streptomycin (4 g/L in drinking water provided six days prior to contractures) Streptomycin (1 mM in vitro) GdCl_3_ (10 μM)	SACs	Streptomycin blocked stretch-induced Na^+^ depolarizations. The effects of GdCl_3_ and streptomycin were qualitatively similar.	[129]
2000	Muscle (Tibialis Anterior)	Rat (Sprague Dawley; 3 months old) (Fisher 344:BN F1 hybrids; 32 months old)	Electrical Eccentric Contraction	Streptomycin (4 g/L in drinking water provided six days prior to contractures) GdCl_3_ (10 μM)	SACs	Stretch-induced NaCl depolarizations in aged skeletal muscle are less sensitive to streptomycin due to the presence of non-specific leakage. The effects of GdCl_3_ and streptomycin were qualitatively similar.	[130]
2003	Muscle (Tibialis Anterior)	Rat (Sprague-Dawley)	Electrical Eccentric Contraction	Streptomycin (4 g/L in drinking water provided six days prior to contractures) GdCl_3_ (10 μM)	SACs	Streptomycin blocked contraction-induced hypertrophy but not increases in contraction force generation. The effects of GdCl_3_ and streptomycin were qualitatively similar.	[30]
2003	Cancer SiHa Cervical Cancer Cells (Epithelial)	Human (Homo sapiens)	Hypotonic Swelling	Streptomycin (10–1000 μM) Gentamicin (10–1000 μM) Netilmicin (10–1000 μM)	SACs	Streptomycin disrupted hypotonicity-induced Ca^2+^ influx but did not influence basal Ca^2+^ level. Hypotonic cell swelling-induced Ca^2+^ influx, in turn, activated a compensatory osmoregulatory Cl^-^ conductance. IC_50_ inhibit initial peak [Ca^2+^]_i_ (μM): Streptomycin (25) Gentamicin (90) Netilmicin (250)	[131]
2003	Muscle (Flexor Digitorum Brevis)	Mouse (*mdx* and C57 BL/10 ScSn (wt))	Electrical Eccentric Contraction	Streptomycin (100 μM) GdCl_3_ (20 μM)	SACs TRPCs	Streptomycin abolished Na^+^ depolarization, reduced force and muscle damage after electrical contracture. Smaller effect size by streptomycin observed in wild-type muscle. TRPCs were cited to represent the aberrantly regulated SACs in dystrophic muscle leading to elevated resting Ca^2+^ entry. The effects of GdCl_3_ and streptomycin were qualitatively similar, suggesting that the spontaneous SAC activity apparent in *mdx* muscle is silent in wild-type muscle.	[132]
2003	Muscle (Flexor Digitorum Brevis (Mouse), Extensor Digitorum Longus (Rat))	Rat and Mouse (Sprague Dawley)	Electrical Eccentric Contraction	Streptomycin (100 μM) GdCl_3_ (20 μM)	SACs	Streptomycin abolished Na^+1^ depolarization following electrical contracture. Similar to streptomycin, GdCl_3_ did not change resting [Na^+^]_i_ but prevented its rise in response to eccentric contractions.	[133]
2005	Muscle (Tibialis Anterior)	Rat (Sprague Dawley)	Mechanical Stretch (40 stretches by ankle rotation)	Streptomycin (300 mg/kg/day; IP 3X/day) EDTA (150 mg/kg; IP 20 min before and 24 h after injury protocol)	SACs	Streptomycin significantly reduced stretch-induced muscle damage. Streptomycin produced greater protection than EDTA administration.	[134]
2005	Heart (Ventricular Myocytes)	Swine (Sus Scrofa Domesticus)	Baseball Impact (40 mph) (*Commotio cordis*)	Streptomycin (2 g intramuscularly; ~115 μM serum levels)	SACs	Streptomycin did not alter the frequency of ventricular fibrillations, but reduced the amplitude of the ventricular ST segment upon impact.	[135]
2005	Muscle (Flexor Digitorum Brevis, Extensor Digitorum Longus)	Mouse (*mdx* and C57 BL/10 ScSn (wt))	Electrical Eccentric Contraction	Streptomycin Patch electrode: 100–200 μM Drinking water: 4 g/L water (3 mM) GsMTx4 (10 μM) GdCl_3_ (20 μM)	SACs	Streptomycin blocked stretch-induced Ca^2+^ increases measured with Patch Clamp recordings and reduced decline in tetanic force following stretch. Patch Clamp recordings revealed SAC blocked by GsMTx4 in both wild-type and *mdx* myotubes. Streptomycin added to drinking water reduced muscle pathology and incidence of central nuclei, in *mdx* muscle. GsMTx4, GdCl_3_, and extracellular Ca^2+^ removal recapitulated the streptomycin-associated reductions in stretch-induced Ca^2+^ increase and muscle force production following tetanic stretch.	[136]
2005	Muscle	Mouse cell line (C2C12)	Myogenic Differentiation	Streptomycin (500–5000 μM) GsMTx4 (0.5 μM) GdCl_3_ (10–100 μM)	SACs	Streptomycin blocks myogenic differentiation. The effect of streptomycin was recapitulated by GsMTx4 and GdCl_3_. GsMTx4 was the least effective and GdCl_3_ was the most effective.	[31]
2009	Fibroblasts (Embryonic Lung)	Human (Homo sapiens)	Cell Migration	Streptomycin (200 μM) EGTA (5 mM) MgCl_2_ (10 mM) Growth Factor: PDGF	TRPM7 SACs	Streptomycin reduced lamella Ca^2+^ flickers and the turning of cells. Ca^2+^ flickering is greatest on leading lamella of migrating cells. EGTA and Mg^2+^ also abolished flickers, implicating channel-mediated Ca^2+^ entry. Lamellar turning requires TRP channel-mediated Ca^2+^ entry and can be activated by platelet-derived growth factor (PDGF). TRPM7 knock-out abolished flickers.	[137]
2005	Muscle	Mouse (*mdx* and C57 BL/10 ScSn (wt)) Human (Homo sapiens)	General Review with Emphasis on Muscular Dystrophy	Streptomycin (100–200 μM) GdCl_3_ GsMTx4	TRPC1	Stretch-induced damage and Ca^2+^ overload are more common in X-linked muscular dystrophies. Streptomycin inhibits stretch-induced Na^+^/Ca^2+^ rise, dye uptake, and muscle damage. The other most common stretch-activated channel blockers (GdCl_3_ and GsMTx4) were highlighted to produce a similar protection as streptomycin.	[138]
2006	Neurons (Spinal Cord Neurons)	Amphibian (Xenopus laevis)	Neurite Outgrowth	Streptomycin, Gentamicin (100–200 μM) GsMTx4 (5 μM) GdCl_3_ (100 μM) SKF-96365 (3 μM) 2-APB (100 μM) Ruthenium Red (1 μM) Ca^2+^ Chelators: BAPTA (5 μm) EGTA (5–25 μM)	TRPCs TRPVs	Blocking stretch-activated channel activity accelerates neurite outgrowth. Gentamicin and SAC/TRP channel blockers (GdCl_3_, GsMTx4, Ruthenium Red, 2APB) slowed neurite outgrowth and reduced growth Ca^2+^ transients (2APB). Removing streptomycin and gentamicin from bathing increased spontaneous Ca^2+^ transient activity in growth cones. Specific Ca^2+^ chelation (BAPTA) slowed neurite outgrowth. A mixture of blockers for voltage-gated Ca^2+^ channels (w-conotoxin (1 μM), nifedipine (100 nM), NiCl (50 μM), w-agatoxin (60 nM), CdCl_2_ (50 μM), SKF-96365 (3 μM)) did not slow neurite outgrowth and did not inhibit growth cone Ca^2+^ transients in response to hypoosmotic swelling.	[139]
2006	Muscle (Flexor Digitorum Brevis)	Mouse (*mdx* and C57 BL/10 ScSn (wt))	Exercise of Intact Mice Single-Channel Patch Clamp Recordings from Isolated Muscle Fibers	Streptomycin (200 μM) Ruthenium Red (50 μM) GdCl_3_ (50 μM) Growth Factor: IGF-1	TRPV2 TRPC1	Streptomycin inhibited single-channel currents on the surface of muscle fibers. Myofibers from dystrophic or exercised mice exhibited higher single-channel activity than unexercised wt fibers. Ruthenium Red and GdCl_3_ similarly inhibited single-channel currents. Insulin-like Growth Factor 1 (IGF-1) increased stretch-activated channel activity in wild-type, but not *mdx* muscles, reflecting channel deregulation.	[140]
2006	Muscle (Extensor Digitorum Longus)	Mouse (*mdx* and C57 BL/10 ScSn (wt))	Downhill Treadmill Running of Intact Mice Eccentric Contractions of Isolated EDL Muscles	Streptomycin: Mice: IP injection (182 mg/kg; 0.5% volume to body mass; ~500 μM in blood) Isolated Muscles: 200 μM in bathing solution.	TRPC1	Streptomycin reduced the increase in stretch-induced (run/electrode) membrane permeability (dye uptake) and preserved force generation in *mdx* fibers. These effects were less apparent and less sensitive to streptomycin in wt muscle fibers. The effects of streptomycin were effectively recapitulated by GsMTx4, a specific SAC blocker. Authors implicated SACs in stretch-induced membrane disruption in muscular dystrophy.	[141]
2006	Muscle	Mouse	General Review with Emphasis on Muscular Dystrophy	Streptomycin Gentamicin GdCl_3_ GsMTx4	TRPC1	Myoplasmic Ca^2+^ overload is strongly implicated in the pathology of X-linked muscular dystrophies. Streptomycin blocked stretch-induced Na^+^/Ca^2+^ rise, dye uptake, and muscle damage. Other stretch-activated channel blockers (GdCl_3_ and GsMTx4) were highlighted to produce similar protection as streptomycin.	[142]
2007	Neurons	Vertebrates	General Review with Emphasis on Neurite Outgrowth	Gentamicin (200 μM)	TRPCs	Neurite outgrowth is modulated by mechanical input. Blocking SACs accelerates neurite outgrowth. Gentamicin increases neurite outgrowth.	[143]
2007	Muscle	Mouse (*mdx* mouse) General Review	Stretch-Activated Single-Channel Recordings via Patch Electrode	General AGAs Neomycin (~200 μM)	TRPCs TRPVs	Neomycin blocked the high spontaneous activity of SACs in dystrophic muscle. The AGAs and lanthanide (trivalent cations such as GdCl_3_) block SACs by directly plugging the channel pore via a process known as open channel block, rather than interacting with the lipid bilayer or via enzymatic means, such as with the phorbol esters or 2-APB. Open channel block exhibits sensitivity to transmembrane voltage.	[93]
2007	Muscle	General Review	SAC Inhibitors	Streptomycin GsMTx-4	TRPC1 TRPC6 TRPA1	Streptomycin, GsMTx-4, and GdCl_3_ all block SAC activity.	[144]
2008	Muscle (Flexor Digitorum Brevis and Extensor Digitorum Longus Muscles) Muscle Cell Line (C2C12)	Mouse (*mdx* and C57 BL/10 ScSn (wt))	Reactive Oxygen Species (ROS)-Induced Ca^2+^ Increments	Streptomycin (200 μM)	TRPC1	Streptomycin blocked ROS-induced Ca^2+^ increase in *mdx*, but not in wild-type muscle fibers. ROS scavengers reduced stretch-induced Ca^2+^ increase. ROS increased Ca^2+^ influx only in muscle cells expressing TRPC1. TRPC1 expression is higher in *mdx* muscle.	[75]
2011	Muscle (leg musculature)	Mouse (*mdx* and C57 C57BL/10 (wt))	Muscle Degeneration	Streptomycin (4 g/L in drinking water) In utero: upon vaginal plug observed up to 6 months post-birth.	TRPC1	In utero treatment with streptomycin delayed the onset of dystrophic symptoms in the limbs of young *mdx* mice but not dystrophic muscular progression in later life. Long-term treatment improved muscle pathology and improved muscle regeneration in older mice.	[145]
2011	Muscle (Diaphragm, Sternomastoid, biceps brachii, Tibialis Anterior)	Mouse (*mdx* and C57BL/10ScCr)	Muscle Degeneration in Response to Exercise	Streptomycin (182 mg/kg bodyweight; IP injections for 18 days)	TRPC1	Streptomycin reduced serum creatine kinase, exercise-induced Ca^2+^ entry & Evans blue dye uptake into the diaphragm and sternomastoid muscles of *mdx* mice.	[146]
2012	Muscle (Extensor Digitorum Longus)	Mouse (Balb/c C57BL/6J (wt) and C57BL/6J TRPC1 knock out)	Electrical Eccentric Contraction	Streptomycin (200 μM) Administered to muscle 30 min before tetanic stimulation.	TRPC1	Streptomycin prevented loss of force after an electrical eccentric contraction in wild-type, but not TRPC1 knock-out muscle. Eccentric contraction-induced loss of desmin, titin, and dystrophin were prevented by streptomycin and TRPC1 knock down.	[147]
2013	Neurons (Spinal Cord Neurons)	Amphibian (Xenopus laevis)	Neurite Outgrowth	Gentamicin (100–200 μg/mL) GdCl_3_ (100 μM) GsMTx (10 μM)	TRPC1	Blocking SACs accelerates neurite outgrowth. TRPC1 knock down precluded effect of gentamicin on axonal outgrowth and Ca^2+^ transients. Effects of gentamicin were observed within 10 s of seconds of addition to the bath. GdCl_3_ and GsMTx recapitulated some effects of gentamicin on axonal outgrowth.	[148]
2019	Muscle Muscle Cell Line (C2C12)	Mouse	Brief (10 min) Pulsing Magnetic Field Exposure (1 mT) (Myogenic Differentiation)	Streptomycin (100 μg/mL) Neomycin (50 μg/mL) Gentamycin (50 μg/mL) Penicillin/Streptomycin (100 U/mL & 100 μg/mL, respectively). SKF-96365 (50 μM) 2-APB (10 μM) Ruthenium Red (10 μM)	TRPC1	Streptomycin, neomycin, gentamycin, and penicillin/streptomycin blocked magnetic induction of proliferation when added shortly before exposure, whereas removal of AGAs a few minutes before exposure did not. Penicillin/streptomycin blocked magnetic induction of Ca^2+^ entry, ROS production, and TRPC1 gene expression. Penicillin/streptomycin blocked magnetic induction of myogenic differentiation. The TRP channel blockers, SKF-96365, 2-APB, and Ruthenium Red and magnetic shielding emulated the effects of AGA treatment across several outcome measures. Magnetic sensitivity most closely followed TRPC1 expression and could be abolished with TRPC1 silencing, but not TRPM8 silencing. Authors deduced that AGAs interfere with mitohormetic adaptations within skeletal muscle cells (Figure 3).	[34]
2021	Stem Cells (Dental Pulp)	Human (Primary Dental Pulp Stem Cells)	Brief (10 min) Pulsing Magnetic Field Exposure (2 mT) Neurogenic Differentiation	Penicillin and Streptomycin (1%)	TRPC1	Penicillin and streptomycin suppressed human dental pulp stem cell proliferation. Magnetic sensitivity followed TRPC1 expression.	[68]
2022	Muscle	Mouse Muscle Cell Line (C2C12)	Brief (10 min) Pulsing Magnetic Field Exposure (1.5 mT)	Streptomycin (100 μg/mL)	TRPC1	Streptomycin blocked magnetic activation of the myogenic secretome response. Streptomycin inhibited TRPC1 protein expression.	[53]
2024	Muscle	Mouse Muscle Cell Line (C2C12)	Brief (10 min) Pulsing Magnetic Field Exposure (1.5 mT)	Gentamicin (50 μg/mL)	TRPC1	Gentamicin prevented magnetically induced myogenic secretome from slowing breast cancer cell migration.	[52,53]
2024	Muscle	Mouse Muscle Cell Line (C2C12)	Brief (10 min) Pulsing Magnetic Field Exposure (1.5 mT)	Streptomycin (100 μg/mL)	TRPC1	Streptomycin prevented combined magnetic field- and light-induced myogenic proliferation when applied during, but not after, exposure. Streptomycin was either applied 15 min prior to or after photo–electromagnetic exposure.	[43]

Abbreviations: 2-Aminoethyl diphenylborinate (2-APB); aminoglycoside antibiotic (AGA); IP: intraperitoneal; SACs: stretch-activated channels; wt: wild type.

The same calcium-sensitive (calcineurin/NFAT), mitochondrial-regulating, transcriptional pathway (PGC-1α) that is activated by TRPC1 [34,114,149] (Figure 3) is downregulated in diabetes [150]. Accordingly, TRPC1/PGC-1 α-dependent oxidative muscle development [34,114,149] is depressed in diabetes [150]. Inhibiting TRPC1 activity hence holds the potential to disrupt metabolism by attenuating oxidative muscle development and activity. On the transcriptional level, the brief activation of TRPC1 by low-energy magnetic fields [34] was capable of provoking the nuclear translocation of the NFATC1 isoform responsible for oxidative muscle determination [151]. On the systemic level, muscular modulation of the whole-body metabolism is conveyed via its secretome. Of relevance is the fact that AGAs have been shown to inhibit TRPC1-mediated growth factor release [53], as well as blocking the actions of growth factors released by TRPC1 activation [52], which could have profound metabolic implications. AGAs may thus potentially disrupt muscle–adipose paracrine crosstalk and systemic metabolism.

## 6. Myogenesis

TRPC1 function is critical for both in vitro [34,101,152,153,154,155] and in vivo [115,116] myogeneses. Muscle phenotype determination is exquisitely responsive to the form of mechanical loading. Prolonged mechanical loading, such as that produced by supporting one’s weight under the force of gravity or during endurance exercise, favors the oxidative phenotype of skeletal muscle [113]. Notably, TRPC1 is essential for muscle’s adaptation to sustained mechanical loading and accordingly, for oxidative muscle development [116,156]. Oxidative muscle is distinguished from glycolytic muscle by its predilection for fatty acids as substrates for mitochondrial energy production, instead of carbohydrates [157,158,159,160,161]. By virtue of its elevated mitochondrial content and associated mitohormetic adaptations [29], the secretome of oxidative muscle is particularly anti-inflammatory [37,38,162]. Oxidative muscle is hence imperative for systemic metabolic balance and resilience to disease. It was previously shown that streptomycin selectively blocked oxidative muscle determination in vitro [33,34,163] and in vivo [146], in alignment with the requirement for TRPC1-mediated calcium entry for oxidative muscle development that reinforces TRPC1 expression [115,116,164]. Streptomycin was also shown to impede TRPC1-mediated muscle secretome release [53] and action [52] in vitro. This same AGA antagonism of skeletal muscle mitohormetic responses would likely be manifested in vivo, undermining the ability of exercise, or other relevant biophysical stimuli, to positively modulate systemic metabolism and health.

Sarcopenia describes the condition of age-related muscle loss that is linked to mitochondrial dysfunction downstream of accruing oxidative stress and chronic inflammation [165,166]. Sarcopenia is characterized by a slow-to-fast muscle fiber shift [167] and attenuated myokine response [168]. By contrast, TRPC1-mediated oxidative enhancement of skeletal muscle is characterized by a fast-to-slow muscle fiber shift (Figure 3) and an anti-inflammatory secretome response (Figure 2). As streptomycin was shown to inhibit oxidative muscle development [33,34], AGAs, in principle, may aggravate the progression of sarcopenia and frailty. Finally, progressive alterations in the cellular mechanical environment have also been proposed to underlie the slowing in muscle development with advancing age. Specifically, a connection between age-dependent changes in the extracellular matrix and mitochondrial dysfunction is being explored in association with sarcopenia [169]. Given that TRPC1 subserves some forms of mechanotransduction, the systemic presence of AGAs may aggravate the regenerative deficits associated with unfavorable alterations in the mechanical environment that may contribute to the development of sarcopenia.

## 7. Adipogenesis

The excessive storage of triglycerides by white adipose tissue is associated with systemic metabolic dysfunction [170] that can be corrected via adipose’s interaction with the muscle secretome [1,2,3]. Critically, AGAs hold the potential to interrupt this beneficial muscle–adipose crosstalk. Non-shivering thermogenesis is the cornerstone of white adipose adaptations (browning or beiging), contributing to systemic metabolic improvement. The induction of non-shivering thermogenesis in white adipocytes relies on the enhanced expression of the mitochondrial protein, uncoupled protein 1 (UCP-1) [171], which is upregulated upon exposure to the muscle secretome [172]. Accordingly, it was previously shown that muscle mitochondrial activation using brief magnetic field exposure conferred a thermogenic effect over white, but not brown, adipose tissue, in association with a significant upregulation of UCP-1 in white adipose [114].

Adipocytes also exhibit AGA-dependent metabolic disruption. However, the antagonism was attributed to a TRP family member other than TRPC1. A comprehensive expression profile analysis of all known 27 human TRP genes was undertaken during white or brown adipogenesis derived from human progenitor cells of either bone marrow or a stromal vascular fraction of subcutaneous adipose origins [39]. The induction of adipose browning (UCP1 expression) was most strongly correlated with two TRP channels, TRPM8 (cold temperature-responsive) and TRPP3 (pH-responsive), in adipose tissue differentiated from either progenitor cell class. Adenylate cyclase modulated the activity of either channel, linking their activity back to mitochondrial respiratory efficiency [39]. The induction of thermogenesis in these mesenchymal cell-derived adipocytes was found to be exquisitely sensitive to the presence of streptomycin. In this context, AGA antagonism of TRPM8 was shown to interrupt adipocyte thermogenesis. Menthol is an agent that chemically recreates the sensation of low temperatures and is a known activator of TRPM8 [173] but not of TRPC1. Streptomycin was capable of competitively inhibiting menthol-activated TRPM8-mediated calcium influx and interfering with the browning of white adipocytes. As TRPC1 expression was equally elevated throughout all stages of adipogenesis (progenitor cell, white and brown adipocytes), it was presumably not required for the induction of uncoupled respiration per se, and its antagonism by streptomycin was inconsequential to adipose browning. Other collateral consequences of AGA-TRPC1 antagonism were not pursued.

AGA antagonism of adipose browning could hence play out on two fronts, systemically (muscle) and locally (adipose). First, the AGAs would attenuate systemic muscle–adipose paracrine crosstalk via the antagonism of TRPC1 on muscle and second, prevent white adipose browning via the antagonism of TRPM8 on white adipocytes. The metabolic disruption associated with AGAs would hence be double-edged with reference to our most inflammatory tissue, adipose.

## 8. In Vivo Antagonism of Mechanotransduction by AGAs

Exercise on the organismal level is fundamentally mechanotransduction on the cellular level. Indeed, mechanotransduction may be a unifying feature of most TRP channels [174,175]. Both stretch-activated channels [93,176,177] and TRPC1 [34] are expressed at their highest levels in muscles during early development. AGAs can hence interfere with muscle development in response to mechanical stimulation. Accordingly, muscle adaptations to exercise have been shown to be interrupted by AGAs (Table 1). Although mechanosensitive channel-mediated calcium entry is necessary for skeletal muscle development and exercise adaptations, deregulated channel activity leading to calcium overload may result in muscle degeneration. Such a deregulation of stretch-activated calcium channels has been implicated in Duchenne and Becker X-linked muscular dystrophies [178,179]. In these conditions, a disruption in normal channel gating is observed, leading to abnormal calcium entry, elevated sarcoplasmic calcium levels, and consequent calcium-dependent necrosis. Evidence exists that the deregulated stretch-activated channel detected in *mdx* muscle has TRP channel origins [174,175], with emphasis on the TRPC family [65], particularly the TRPC1 subunit of the channel [92,138,180].

Streptomycin use is compatible with in vivo studies examining mechanosensitive processes due to its ability to be administered to animals in their drinking water without severe toxicity. Given its noted antagonism over TRPC1, streptomycin has been employed in numerous studies to attenuate either elevated basal sarcoplasmic calcium entry into dystrophic muscle or that stimulated in response to exaggerated muscle contractures (Table 1). Streptomycin was shown to directly block a TRP channel species in skeletal muscle fibers isolated from *mdx* mice, a mouse model for human X-linked muscular dystrophy [140], as well as from myotubes grown from *mdx* muscle biopsies [136]. Acutely applied streptomycin reduced the eccentric contraction-induced calcium entry and associated muscle damage of isolated *mdx* extensor digitorum longus (EDL) muscles [136]. Streptomycin also reduced force loss after the onset of eccentric contractures in wild-type muscle fibers, but not in the EDLs of TRPC1 knock-out mice [147], supporting the hypothesis that the demonstrated stretch-activated channels are synonymous with TRPC1. An analogous effect was observed in normal rat tibialis anterior muscles, whereby acutely applied streptomycin reduced eccentric contracture-induced muscle damage [45]. Previous animal trials have also shown that systemically administered AGAs protected against muscle damage in dystrophic mice by specifically targeting TRPC1 [141,145,146] or generic stretch-activated channel activity [136], which evidence indicates represents the same channel complex [92,138]. Streptomycin in these instances may have protective physiological consequences with reference to skeletal muscle calcium overload and resultant myonecrosis.

An additional protective mechanism for gentamicin has also been described in *mdx* mice that is attributed to its ability to suppress the premature dystrophin stop codon, restoring dystrophin expression [181]. Consequently, gentamicin was employed in a human clinical trial of Duchenne muscular dystrophy (DMD) with modest success [182]. While no change in dystrophin expression was observed following the gentamicin treatment in DMD patients, reductions in creatine kinase leakage into the serum from muscle were observed, indicating improved muscle membrane integrity, resembling preclinical results from *mdx* mice administered streptomycin [136] (also see Table 2). Given that gentamicin, like streptomycin, has been shown to block stretch-activated/TRPC1 channels in diverse cell types [34,44,91,126,131,139], it would be important to determine the relative contribution of each of these protective mechanisms in dystrophic muscle [141,142].

Developmentally troubling is the widespread use of antibiotics during pregnancy [183]. In infants and the young, the antagonism of signal transduction by AGAs would not be entirely determinant of developmental alterations but would contribute to biosynthetic and microbiomal disruptions of developmental consequence common to most antibiotic classes. Of special concern in the context of the present discussion is the common use of gentamicin in neonatal and pediatric clinical scenarios where it is routinely administered intravenously or intramuscularly [184]. Interestingly, the heart physiology appears to be less sensitive to streptomycin application (Table 3), which may be related to its relatively dampened hypertrophic response to physiological levels of stretch (mechanical stimulation).

## 9. Non-TRP Channel Classes Shown to Be Sensitive to AGAs

Promiscuity is a feature that plagues most ion channel antagonists and, in this respect, AGAs are no exception. AGAs also have been shown to block other calcium channel classes but with a lower affinity than for TRPC1 [128,141]. For instance, voltage-gated calcium channels (L-type) are also blocked by AGAs in diverse species and cell classes [120,191,192]. Notably, the affinity of AGAs for the L-type voltage-gated calcium channel pore [120,192,193] is lower than that for the pore region of TRPC/SAC channels [91], binding in the mM range rather than the μM range, respectively [128,141]. Streptomycin also has been shown to block the mechanosensitive Ca^2+^-permeable channel, Piezo1, in a use-dependent manner [194]. Streptomycin was shown to bind to an open conformation of the Piezo1 channel in the mM range. Importantly, Piezo1 has been recently shown to contribute to muscle regeneration [195] and maintenance [196]. Finally, AGAs have also been shown to “plug” the Maxi-K^+^ and Ca^2+^-activated K^+^ channels in the low mM range [197,198]. These other (non-TRP) channel classes, due to their lower sensitivity to AGAs, would contribute comparatively less to systemic AGA-induced metabolic disruption [128].

## 10. AGA Antagonism of Other TRP Channels

Certain TRP channels have also been implicated in the transduction of sound and body position relative to gravity originating in the saccular and vestibular organs, respectively [199,200], in their capacity as acoustic vibration and shear stress sensors, respectively [124,126]. These TRP channels include TRPV4, TRPN1, TRPA1, and TRPML3 [199,200,201] and are essentially mechanosensory modalities. Accordingly, AGAs are also implicated in ototoxicity [202], as well as having been directly shown to block vibrational mechanosensitive gating via an open channel-dependent binding of AGAs to the channel’s cation conduction pathway [124,126].

TRPV channels in a variety of tissues are also susceptible to AGA antagonism. In sensory neurons, Ca^2+^ entry via TRPV1 channels was blocked by AGAs [203], exhibiting the highest sensitivity to neomycin and streptomycin (IC50 ~400 nM) and the least sensitivity to gentamicin. In the nervous system, TRPV1 is necessary for the perception of heat and pain. The TRPV1 and TRPV4 channels have also been shown to play a decisive role in vascular muscular remodeling [204], wherein AGA-mediated TRPV1 antagonism may be ultimately found to have critical ramifications. In skeletal muscle cells, however, magnetotransduction was unaffected by TRPV4 antagonism with Ruthenium Red (10 μM), and TRPV1 developmental expression was induced after myoblast expansion when magnetosensitivity was greatest [34], whereas TRPC1 expression was found to be necessary and sufficient to endow magnetoreception in myoblasts [67]. On the other hand, gentamicin has been shown to permeate TRPV1 [205,206,207] and TRPV4 [207,208] channels in various cell types. The TRPM7 and TRPM8 channels are blocked by streptomycin in human embryonic lung fibroblasts [137] and human adipocytes [39], respectively.

This review purposefully focused on the acute blockage of the TRPC1 and TRPM8 channels’ conduction pores by AGAs, particularly with regards to the muscle–adipose crosstalk (Figure 1). The more generalized and protracted roles of AGAs over cellular protein synthesis [42], antimicrobial resistance [209] and the microbiome [210] were not explored, despite their importance to human physiology. Of note, the knock out of the TRPV1 and TRPA1 channels, independently and combined, previously implicated in cellular responses antagonized by AGAs (see above), led to a form of systemic dysbiosis in mice that was associated with higher lipid biosynthesis [211]. Sensory neurons innervating the gut were proposed to be the source of the implicated TRPA1 and TRPV1 channels, presumably modulating the microbiome by influencing the permeability of the gut epithelial lining. It may be thus inferred that systemic AGA administration may similarly contribute to lipid accretion and a predisposition to systemic inflammation. This provocative scenario merits its own dedicated review, as it implicates both antimicrobial resistance and antagonism towards TRP channel function as a result of the presence of AGAs.

## 11. Conclusions

TRP channels are broadly implicated in tissue determination and development. TRPC1 is the founding member of the TRPC subfamily and, for that matter, the entire TRP superfamily. TRPC1 is the most ubiquitously expressed of all TRP channel subunits, underscoring its importance to human physiology. Ample evidence implicates TRPC1 expression in the transduction of two unyielding biophysical stimuli of developmental importance, gravitational force (mechanical loading) and magnetism. An underappreciated feature of AGAs is their ability to impede Ca^2+^ permeation through various TRP channel complexes, including TRPC1- and TRPM8-containing channels. The use of AGAs during in vitro or in vivo studies will hence alter the natural course of tissue development and may lead to erroneous interpretations of the data generated therein. The disruption by AGAs of TRPC1-mediated oxidative muscle development could have profound repercussions for systemic inflammation, metabolic balance, frailty, and aging. Streptomycin has also been shown to cause another form of metabolic disruption by blocking Ca^2+^ permeation via the thermosensitive TRPM8 channel. AGA antagonism of TRPM8 in white adipocytes prevented the induction of uncoupled respiration and non-shivering thermogenesis, which effectively entrenched the white phenotype. This effect would facilitate the establishment of systemic inflammation, as well as interfering with the metabolically beneficial muscle–adipose crosstalk that is dependent on the conversion of white to beige adipocytes. The combined antagonism of TRPC1 and TRPM8 channels by systemic AGA administration could thus have profound consequences on the whole-body metabolic balance by first, compromising mitochondrial (mitohormetic) adaptations at the cellular level and second, by attenuating muscle–fat paracrine crosstalk at the systemic level. The implications of AGA antagonism on other TRP channel complexes predominantly expressed in other tissues remain to be elucidated but may be equally critical. Given the broad developmental implications of the TRP channel family and the described antagonism of several TRP channel species by AGAs, prudence must be exercised when employing these antibiotics during in vitro or in vivo developmental studies.

## Figures and Tables

**Figure 1 cells-13-01273-f001:**
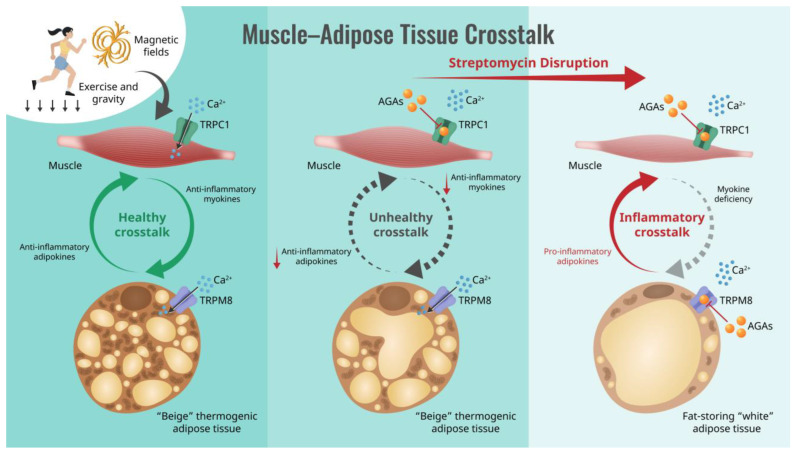
Schematic of how the aminoglycoside antibiotics (AGAs) interrupt muscle–adipose crosstalk by inhibiting Ca^2+^ entry via transient receptor potential (TRP) cation channel classes, causing systemic metabolic disruption. Semicircular arrows indicate the direction of muscle-adipose paracrine crosstalk ranging between anti-inflammatory (green; left) to pro-inflammatory (red; right) character.

**Figure 2 cells-13-01273-f002:**
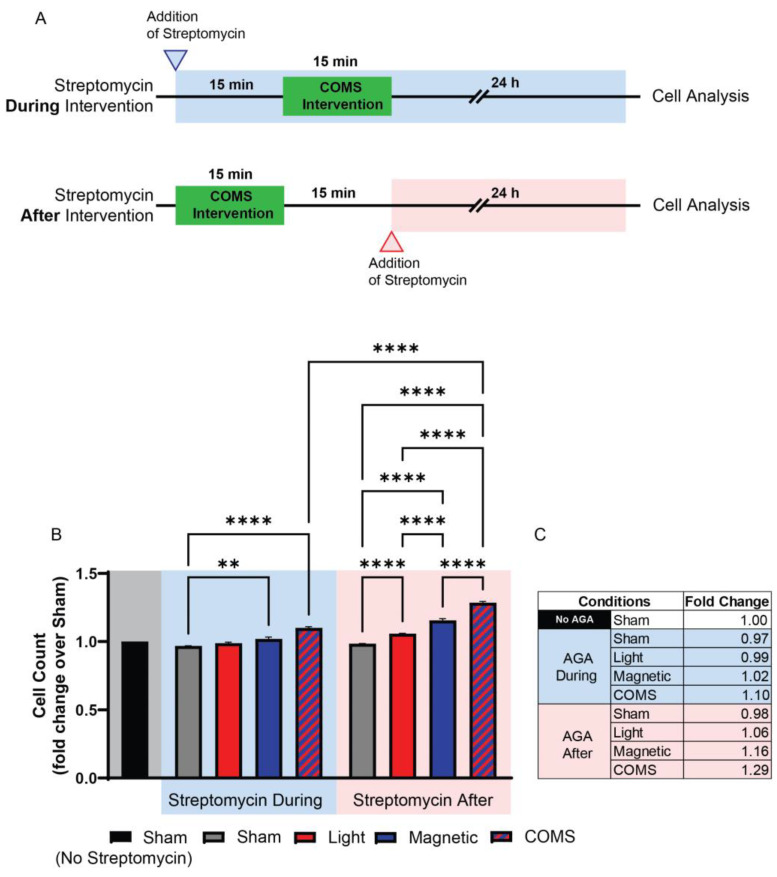
Streptomycin present during exposure suffices to preclude developmental responses to biophysical stimuli. Streptomycin (100 μg/mL) added to the bathing media 15 min before, but not 15 min after, exposure to light (optical), magnetic fields, or concurrent optical and magnetic field stimulation (COMS), inhibited the typical stimulation of myoblast proliferation. (**A**) Experimental timeline depicting when streptomycin was administered in the “During” (top/blue) or “After” (below/red) scenarios with reference to electromagnetic exposure (green shaded region). (**B**) Murine myoblast proliferation as aforementioned, with streptomycin added to the bathing media (100 μg/mL) either 15 min before (blue-shaded region) or after (red-shaded region) exposure to the different exposure paradigms. The shaded areas indicate either the absence of streptomycin (gray) or its application before and during (blue) or after exposure (red) to the indicated conditions. (**C**) Tabulated fold changes of live cell number for the different exposure conditions relative to sham (no biophysical exposure, no streptomycin). In summary, a criterion for streptomycin antagonism of electromagnetic regenerative responses is that it be present at the time of exposure. AGA: aminoglycoside antibiotic. Data were analysed using one-way ANOVA followed by multiple comparison tests. Significance levels are indicated as follows: ** *p* < 0.01, and **** *p* < 0.0001. The error bars represent the standard error of the mean. Adapted from Iversen et al. 2024 [43].

**Figure 3 cells-13-01273-f003:**
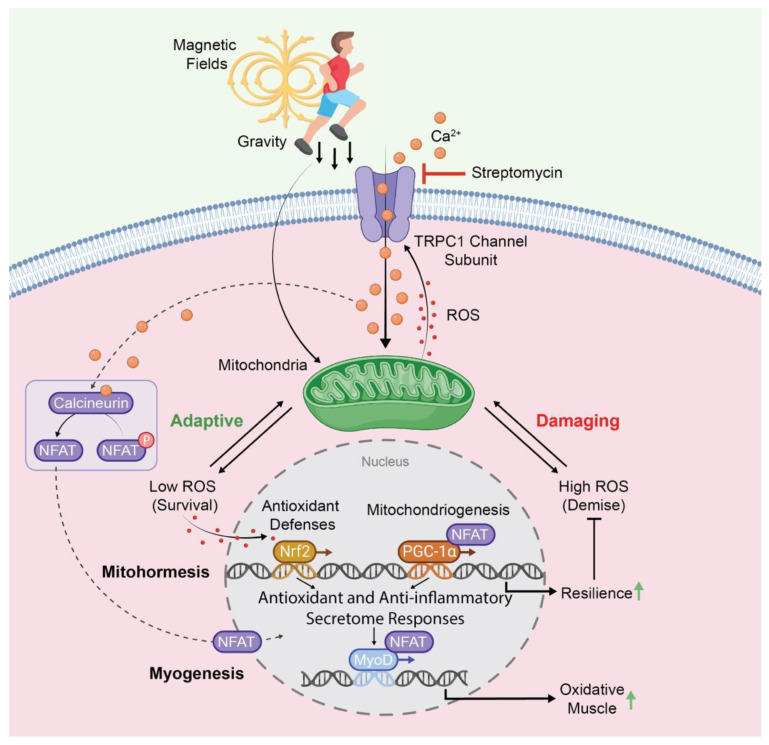
TRPC1-dependent transcriptional and enzymatic pathways co-activated by gravity-mediated (mechanical loading) or electromagnetic stimuli impinging on muscle. The orange circles represent Ca^2+^ and the red dots represent ROS. TRPC1 activation enhances mitochondrial biogenesis and myogenesis towards the oxidative (slow) phenotype by contributing to the mitohormetic adaptations associated with resistance to oxidative stress. The TRPC1-mediated activation of these same pathways should forestall the development of sarcopenia and be reversed by aminoglycoside antibiotics, such as streptomycin. Abbreviations: ROS: reactive oxygen species; NFAT: Nuclear Factor of Activated T-Cells; PGC-1α: Peroxisome Proliferator-Activated Receptor Gamma Coactivator 1-alpha; Nrf2: Nuclear Factor Erythroid 2-Related Factor 2; MyoD: Myoblast Determination Protein 1. For further details, kindly refer to references [34,96].

**Table 3 cells-13-01273-t003:** AGA antagonism of heart stretch-activated channels reveals lower efficacy.

AGA Antagonism of Stretch-Activated Channels in Heart						
Year	Tissue/Cell Type	Animal	Context	AGA/Channel Modulator (Dose)	Findings	Ref.
1998	Heart (Neonatal Cardiomyocytes)	Rat (Wistar)	Stretch–Induced Cardiomyocyte Hypertrophy	Streptomycin (5 × 10^−4^ mol/L; 60 min pretreatment to stretch) GdCl_3_ (10^−5^ mol/L; 60 min pretreatment to stretch)	Streptomycin did not activate MAP kinase. GdCl_3_ did not activate MAP kinase.	[185]
2003	Heart (Ventricle)	Guinea Pig (Cavia porcellus)	Axial stretch	Streptomycin (40 μM) BAPTA-AM (5 μM)	Streptomycin prevented stretch-induced increase in action potential duration. Streptomycin precluded registry of stretch-induced cation currents. BAPTA (Ca^2+^ chelator) also abolished prolongation of action potential.	[186]
2005	Heart (Ventricular Myocytes)	Swine (Sus Scrofa Domesticus)	Baseball Impact (40 mph) (*Commotio cordis*)	Streptomycin (2 g intramuscularly; ~115 μM serum levels)	Streptomycin did not alter the frequency of ventricular fibrillations, but reduced the amplitude of the ventricular ST segment upon impact, implicating SACs in this aspect of the response.	[135]
2005	Heart (Sino-Atrial Node) Strips and Cells)	General Review (Stretch Effects on Sino-Atrial Node Pacemaking)	SAC Activation	Streptomycin (40–500 μM)	Right atrium dilation increases heart rate and chronotropy. SACs proposed to underlie chronotropic response. Streptomycin does not alter the chronotropic response.	[187]
2007	Heart (Myocardial Infarction)	Rat (Wistar)	Ischemic Myocardium. Stretch by Balloon Inflation	Streptomycin (200 mmol/L)	Streptomycin significantly inhibited the occurrence of arrhythmias. Proposed streptomycin blocked SACs.	[188]
2008	Heart (Engineered Heart Muscle)	Rat (F344)	Stretch (1 Hz, 10% strain for 7 days continuously)	Streptomycin (not given)	Streptomycin prevents the engineered heart muscle from adapting to stretch.	[189]
2008	Heart (Right Ventricular Trabeculae)	Mouse (Swiss White, C57BL/10ScSn (wt), or *mdx*)	Stretch-induced	Streptomycin (400 μM) GdCl_3_ (10 μM) GsMTx-4 (10 μM)	Streptomycin reduced slow force response associated Ca^2+^ influx. Resting Ca^2+^ entry was higher in myocytes from old *mdx* mice, which was blocked by SAC blockers (streptomycin, GdCl_3_, GsMTx-4) and associated with an elevated expression of TRPC1.	[190]
2011	Heart	Mouse (*mdx*)		Streptomycin (Chronic) (4 g/L water, in utero)	Long-term treatment with streptomycin protected against limb muscle pathology, reduced fibrosis, increased sarcolemma stability, and promoted muscle regeneration in older mice. However, streptomycin treatment did not show positive effects in the diaphragm or heart muscle, and heart pathology was even worsened.	[145]

Abbreviations: aminoglycoside antibiotic (AGA); stretch-activated channel (SAC); wt: wild type.

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
