# Peer review of "Are Aminoglycoside Antibiotics TRPing Your Metabolic Switches?"

_cells, 2024, doi:10.3390/cells13151273_

Round 1

Reviewer 1 Report

Comments and Suggestions for Authors

Summary

This review focuses on aminoglycoside antibiotics (AGAs), which have a bactericidal effect and are able to block TRP channels, thereby blocking calcium influx. Aminoglycosides have a broad spectrum of activity and can be used against many infections. An important representative of the AGA group is streptomycin, which in turn is able to block TRP channels. This is the starting point for this review on the role of TRP channels in tissue determination and development. This is an interesting aspect especially with regard to skeletal muscle and adipose tissue. TRP channels are developmentally very old ion channels. The TRPC1 channel (canonical subgroup), which is involved, for example, in the etiology of X-linked muscular dystrophies, is the first of these. TRPC1 is the most ubiquitously expressed of all TRP channels. It is interesting to note in this review that aspects related to gravity (mechanical stress) and magnetism, which are not frequently researched, are also addressed. The authors conclude that there is much to suggest that TRPC1 expression is involved in the transduction of these two mentioned biophysical stimuli of developmental importance. In addition to TRPC1, the menthol receptor TRPM8 was also named as a switching point. The combined antagonism of TRPC1 and TRPM8 by systemic AGA administration may have significant consequences for metabolic balance. The authors further conclude that the effects of AGA antagonism on other TRP channels prevalent in other tissues are critical and should be further elucidated. Due to the wide-ranging effects of the TRP channel family on the described antagonism of several TRP channel types by AGAs, the authors advise caution when using these antibiotics in in vitro or in vivo development studies.

General Comments

This review is generally well written, mostly easy to understand and easy to read. The introduction gives a good overview of the importance of muscle and adipose tissue (paracrine factors, myokines and adipokines). In particular, adipose tissue in the body is not organized in such a way that it can resist gravity and that the activation of muscles through "movement" improves the body's metabolism (interruption of adipose inflammation). The detrimental aspects of AGAs are clearly addressed and well-illustrated (Schematic of how the AGAs interrupt muscle-adipose crosstalk causing systemic metabolic disruption). A table on AGA antagonism of exercise effects (implicated stretch-activated channels) contributes to understanding and a good overview (with references), whereby the importance (adaptation) of mitohormesis is also pointed out (ROS act as signaling molecules to initiate a cascade of cellular events that ultimately protect the cells from harmful effects). I think the illustration in Fig. 1 (Schematic of how the AGAs interrupt Muscle-Adipose crosstalk causing systemic metabolic disruption) is well done. Chapter 2 provides a good explanation of how TRP channels transduce biophysical stimuli into mitormetic responses. The extensive tables also provide a good overview of the current state of research. The aspects relating to gravity and the earth's magnetic field are somewhat more difficult to provide. A few additions regarding associations with AGAs and TRP channels might be useful here. Some medical information would also make it easier to understand medical contexts (e.g. adipogenesis, sarcopenia). Overall, I recommend a (minor) revision.

Specific comments

Background (from line 120): How effectively inhibit AGAs developmental signal transduction during the mentioned biophysical stimuli? And how can the time course of action be very brief?

From line 156 (Fig. 2): It would be good if the number of measurements could be mentioned in the legend text. Which statistics were used. Are they SD or SEM error bars?

From line 167: What exactly are these indications? How exactly is the biophysical signal transduction disturbed by TRP channels? Are there any studies on this?

Chapter 2 (from line 171): It is not clear how the mentioned biophysical stimuli affect TRP channels at the cellular level, especially TRPC channels. What about other TRP channels that can actually be physically activated (temperature, osmosis, UV light). The relationship between ROS (mitochondria), mitohormesis and TRP channel function may be not entirely clear for readers. Perhaps, a simplified illustration may help in this matter.

Chapter 3 (from line 210): AGAs have the ability to block calcium influx via TRP channels. How does this compare to negative controls (2-APB? OAG in TRPC1, PMID: 9930701, PMID 11970863)? Does streptomycin have a voltage-dependent effect? 

Chapter 4 (from line 233): Mechanotransduction channels are Piezo1, Piezo2, TASK1, TREK1, TRPA1, TRPC1, TRPC2, TRPC3, TRPC6, TRPM2, TRPP2, TRPV4 (PMID 24215462, 15909178, 19279663). How can it be explained that the aspects in this review mainly refer to TRPC1 only? The citations [38, 96, 99] have nothing to do with TRPC1. How can this be related to magnetic therapy? Perhaps, a simplified illustration could also help here? Perhaps in combination with chapter 8?

Table 2, page 11, Ref. [131]: It should be noted in the table that these are osmo-sensitive chloride channels (cell volume). Perhaps this addition could also be mentioned in chapter 8?

Chapter 5 (from line 322): It would be good if a few more medical aspects of sarcopenia (a form of muscle atrophy) could be added (pathogenesis? What about inflammatory processes?). The loss of muscle mass as well as the decrease in muscle strength and physical endurance are characteristic of sarcopenia, the degenerative age-related deterioration of skeletal muscles. Why is TRPC1 crucial for muscle adaptation? A short addition to the manuscript text would be helpful.

Chapter 6 (from line 355): A small supplement with medical information and the latest research would also be helpful here. For example, it should be explained that metabolic diseases can develop if the adipocytes are not working properly. See also PMID 38565923. Using this approach, scientists were able to take snapshots of the cells during their developmental process and track the dynamic changes in protein localization during this process.

Chapter 9 (from line 475): The cited study by Raisinghani and Premkumar [199] (line 486) is interesting here, but is not discussed in detail with regard to possible pain reduction. If calcium influx via TRPV1 can be blocked with AGAs such as neomycin and streptomycin, how would this affect muscle-adipose-tissue crosstalk? See PMID 22820913.

Conclusions (from line 491): Combined antagonism of TRPC1 and TRPM8 by systemic administration of AGA could conceivably have profound effects on metabolic balance throughout the body as the authors conclude. As there is still a great need for elucidation, the authors advise caution when using these antibiotics in in vitro or in vivo development studies. This is understandable, but it would be good if the authors would briefly add specific reasons. Are there any studies that take an equally critical view of this?

Minor

I recommend changing the title. In my view, “TRPing” may be not clearly understandable for readers. TRP mean transient receptor potential. I recommend formulating this more precisely.

Comments on the Quality of English Language

I did not notice any linguistic inconsistencies while reading. A minor linguistic revision may still be necessary.

Reviewer 2 Report

Comments and Suggestions for Authors

Comments to the Authors-

The paper is very interesting and the results collected could be very important to the scientific community. However, minor changes are needed. Below are some tips for authors.

 -The text of the manuscript contains many acronyms which make difficult to read, please define acronyms when first using them by alternating them with the long name to make the manuscript more fluid and understandable.

 -The legends of the figure and tables must be modified by defining all the acronyms reported.

Reviewer 3 Report

Comments and Suggestions for Authors

This is a very interest work but it lacks significant mention of the gut.

TRP channels are also present in the gut. Nothing is mentioned about that. The authors are kindly requested to revise their manuscript adding separate paragraphs about the effect of aminoglycoside antibiotics on the gut integrity and their action on the gut microbiome.

It is a very interest paper that needs some serious revision to be appropriate for publication. 

Comments on the Quality of English Language

The English language needs proofreading by a native speaker.
